# Therapeutic Potential of Tumor Metabolic Reprogramming in Triple-Negative Breast Cancer

**DOI:** 10.3390/ijms24086945

**Published:** 2023-04-08

**Authors:** Gyöngyi Munkácsy, Libero Santarpia, Balázs Győrffy

**Affiliations:** 1National Laboratory for Drug Research and Development, Magyar Tudósok Körútja 2, 1117 Budapest, Hungary; 2Oncology Biomarker Research Group, Research Centre for Natural Sciences, Institute of Enzymology, Magyar Tudósok Körútja 2, 1117 Budapest, Hungary; 3Seagen, Dammstrasse 23, 6300 Zug, Switzerland; 4Department of Bioinformatics, Semmelweis University, Tűzoltó u. 5-7, 1094 Budapest, Hungary; 5Department of Pediatrics, Semmelweis University, Tűzoltó u. 5-7, 1094 Budapest, Hungary

**Keywords:** triple-negative breast cancer, cancer metabolism, reprogramming, signaling pathway, tumor microenvironment, therapeutic signaling

## Abstract

Triple-negative breast cancer (TNBC) is the most aggressive subtype of breast cancer, with clinical features of high metastatic potential, susceptibility to relapse, and poor prognosis. TNBC lacks the expression of the estrogen receptor (ER), progesterone receptor (PR), and human epidermal growth factor receptor 2 (HER2). It is characterized by genomic and transcriptional heterogeneity and a tumor microenvironment (TME) with the presence of high levels of stromal tumor-infiltrating lymphocytes (TILs), immunogenicity, and an important immunosuppressive landscape. Recent evidence suggests that metabolic changes in the TME play a key role in molding tumor development by impacting the stromal and immune cell fractions, TME composition, and activation. Hence, a complex inter-talk between metabolic and TME signaling in TNBC exists, highlighting the possibility of uncovering and investigating novel therapeutic targets. A better understanding of the interaction between the TME and tumor cells, and the underlying molecular mechanisms of cell–cell communication signaling, may uncover additional targets for better therapeutic strategies in TNBC treatment. In this review, we aim to discuss the mechanisms in tumor metabolic reprogramming, linking these changes to potential targetable molecular mechanisms to generate new, physical science-inspired clinical translational insights for the cure of TNBC.

## 1. Introduction

The incidence of breast cancer is increasing globally and is a leading cause of cancer-related deaths among women worldwide [1]. Triple-negative breast cancer (TNBC), which accounts for approximately 15–20% of all breast cancer cases, is characterized by the absence of estrogen receptor (ER), progesterone receptor (PR), and human epidermal growth factor receptor 2 (HER2) expression. TNBC is considered to be an aggressive subtype of breast cancer, with a poor prognosis due to limited treatment options and a high rate of relapse [2].

Despite recent advances in breast cancer therapy, the overall survival rates for patients with TNBC remain low [3], highlighting the need for a deeper understanding of the underlying biology of this subtype. The conceptualization of the cancer hallmarks (sustaining proliferative signaling, evading growth suppressors, resisting cell death, enabling replicative immortality, inducing angiogenesis, activating invasion and metastasis, reprogramming energy metabolism, evading immune destruction, deregulating cellular energetics and genome instability and mutation) offers an opportunity to find an effective point of attack against this aggressive type [4]. Among these, a growing body of evidence suggests that dysregulated metabolism is a key characteristic of TNBC, contributing to its metastatic behavior and resistance to therapies [5,6,7].

Understanding the metabolic dysregulations in TNBC and developing therapies that target these pathways may provide new avenues for improving outcomes for patients with this aggressive subtype of breast cancer. This manuscript will provide an overview of the current state of knowledge in this area, including recent references and emerging therapeutic approaches.

## 2. Cells and Interactions in the Tumor Microenvironment

The tumor and the surrounding cells act as a dynamic functional network in the tumor microenvironment (TME). Levels of oxygen, nutrients, and the intermediate products of metabolism, which are essential for the tumor’s survival, growth, and proliferation, are also in continuous transformation [8,9]. Among the connective tissue elements that make up the largest part of the stroma are cancer-associated fibroblasts (CAF), cancer-associated adipocytes (CAA), and tumor-associated monocytes/macrophages (TAMs). These cells play an important role in the maintenance, survival, and metastasis of tumors and tumor cells [10]. The TME contains several non-immune cells such as fibroblasts, adipocytes, vascular smooth muscle and endothelial cells, and immune cells such as T-lymphocytes, macrophages, and natural killer cells [11].

### 2.1. Cancer-Associated Fibroblasts (CAFs)

Most of the tumor stroma is composed of cancer-associated fibroblasts (CAFs), which play an important role in cancer progression through their molecular cooperation with cancer cells. Tumor cells produce chemokines (CXCL2-CXC motif chemokine ligand 2), cytokines (IL-6–interleukin 6), growth factors (TGF-transforming growth factor), and platelet-derived growth factor (PDGF), thereby stimulating the transformation of normal fibroblasts into activated CAFs [12,13]. CAFs regulate tumor growth, metastasis, and angiogenesis by the secretion of transforming growth factor-β (TGF-β), IL-6, CXCL12, and chitinase-3-like-1 (CHI3L1) [14,15]. CAFs induce radio resistance in breast cancer via different paracrine signaling pathways [16]. CAF reprogramming also causes a decrease in the expression of mitochondrial transcription factor A (TFAM), which induces the loss of caveolin 1 (CAV-1) and a decrease in oxidative phosphorylation while glycolysis increases [17] (Figure 1). In glycolytic CAFs, the increased levels of monocarboxylate transporter 4 (MCT4) allow the export of lactate, followed by lactate influx into tumor cells via MCT1, which enters the TCA (tricarboxylic acid) cycle and promotes oxidative metabolism [18]. To maintain the metabolic needs due to tumor proliferation, CAFs also produce different amino acids (such as glutamine and alanine) as other carbon sources, which support the growth of tumor cells [19] (Figure 1).

### 2.2. Adipose Tissue (AT)

The association between increased adipose tissue (AT) and worse prognosis in cancer patients has been known for a long time [20]. Indeed, most adipose tissues contain adipocytes, which can promote the proliferation of cancer cells, angiogenesis and metastasis through different secreted molecules [21] (Figure 2). Signaling pathways related to carcinogenesis can be regulated by adipokines secreted by neighboring fat cells and recognized by specific receptors on the surface of tumor cells [22]. For example, PI3K/AKT (phosphoinositide 3-kinase/protein kinase B), MAPK/ERK (mitogen-activated protein kinase/extracellular signal-regulated kinase) and STAT3 (signal transducer and activator of transcription proteins 3) signaling pathways can be selectively activated by leptin, hepatocyte growth factor (HGF), insulin-like growth factor 1 (IGF-1), and resistin, which promote cell survival and proliferation [23,24]. Leptin, HGF, and VEGF-A (vascular endothelial growth factor A) cells produced by adipocytes also play a role in blood vessel formation [25]. Leptin is proven to have a pro-proliferative effect on cancer cells, while adiponectin acts as a pro-apoptotic factor. Remodeling of the extracellular matrix of the surrounding adjacent tumor area may also be associated with adipocytes, as the release of collagen VI and matrix metalloproteinase 11 (MMP11) promotes cancer invasion [26] (Figure 2). It has been described that adipocytes also promote invasion and drug resistance by inducing CAFs in bone marrow metastases [27].

### 2.3. Tumor-Associated Macrophages (TAMs)

Macrophages constitute an important component of the immune system during tumorigenesis in the early stages and in the adaptive immune response. In the early stage of cancer, the disease is marked by immune rearrangement and inflammation. Tumor cells secrete a cytokine mixture containing TGFβ1, which acts as a strong immunosuppressant by blocking the maturation of monocytes through its CSF-1 (colony-stimulating factor 1) content [28]. In this malignant tumor-oriented microenvironment, tumor-associated macrophages (TAMs) are generated from circulating blood monocytes. After exosmosis, they differentiate into macrophages and become key components of the TME [29].

There are two main different subtypes of TAMs. M1-type macrophages, which are associated with an inflammatory response by releasing proinflammatory cytokines inducing a Th1 (T helper type 1) immune response; and M2-type macrophages, which are usually associated with tumor progression through their secretion of interleukin-10 (IL-10) and TGF-β, inhibiting Th1 immune responses, and by promoting tumor invasion and metastasis [30]. They also secrete different angiogenic factors (e.g., VEGF), through which they may promote tumor angiogenesis and provide nutritional and metastasis signaling for tumor growth.

TAMs can suppress the intervention of other immune cells through IL-10 [31] and arginase I release [32]. The inflammatory environment is caused also by the secretion of reactive oxygen species (ROS) and nitrogen radicals (NOS) [33], which can potentially generate mutations. Some cytokines related to mutagenic events such as TNF-α and macrophage migration inhibitory factor (MIF) can also be released [34]. Paracrine interaction between monocytes/macrophages and cancer cells induces metastasis after the initial processes. One such mechanism involves the release of CSF-1 from tumor cells attracting monocytes to the tumor environment, and the activation of EGF signaling promoting the migration of tumor cells [35]. TAM-derived tumor necrosis factor alpha (TNF-α) is another enhancer of metastatic behavior through the induction of MIF and extracellular matrix metalloproteinase inducer (EMMPRIN). In this milieu, cancer cell-derived IL-4, CXCL12, FGF (fibroblast growth factor), PDGF, and TAM-derived TGF-β [36], MMPs, urokinase-type plasminogen activator (uPA), and IL-6 affect ECM proteolysis, thus promoting the migration of tumor cells and the eventual release of important mediators of tumor spread [37]. CCL18 (chemokine ligand 18) secreted by TAMs are responsible for the development of a chemoresistance-inducing phenotype in breast cancer [38].

### 2.4. Tumor-Infiltrating Lymphocytes (TILs)

The TNBC microenvironment is infiltrated by large amounts of tumor-infiltrating lymphocytes (TILs), including CD3+ T and CD20+ B cells and CD38+/CD138+ plasma cells. CD3+ T lymphocytes have an important role in the establishment of the immune response. This population can be divided into at least three subgroups, as follows: CD8+ T lymphocytes, CD4+ helper T lymphocytes and CD4+ regulatory T cells (Tregs) [11]. CD8+ T lymphocytes are the main driving force of the anti-tumor immune response. After activation by the major histocompatibility complex (MHC), CD8+ cells release interferon-γ (IFN-γ) and become a killer of tumor cells. CD4+ helper T lymphocytes support tumor cell death mediated by CD8+ T lymphocytes, thus participating in the development of the antitumor immune response. CD4+ T cells have two subtypes (Th1 and Th2) according to the secreted cytokines, and both strengthen the immune response. Tregs make up 10% of all CD4+ T lymphocytes in the peripheral blood of healthy people, rising up to 30–50% in the presence of tumor lesions, inhibiting the activation of CD8+ T and CD4+ T lymphocytes and hence, playing an essential role in immunosuppression and angiogenesis [39,40].

### 2.5. Tumor-Associated Neutrophils (TANs)

Neutrophils are the first line of defense on the site of inflammation. These cells interact with several other immune cells, secreting chemokines and peroxidases. Tumor-associated neutrophils (TANs) are very important immunosuppressor components of the TME-inducing antitumor effects, either through cytotoxicity or lysis of the tumor cells directly [41]. N2-type TANs support tumor growth and convert TANs to antitumor N1-type neutrophils by blocking TNF-α or boosting IFN. TANs and TAMs are in close relationship in the TME due to their similar effect on TGF-β [42].

### 2.6. Natural Killer (NK) Cells

In the absence of specific immunity, natural killer (NK) cells are capable of causing tumor cell death. NK cells are produced mainly in the bone marrow, and when they are in touch with tumor cells, they secrete cytokines like cytolytic TNF-α and IFN-γ. In a study, patients with higher NK cell infiltration had a better pathological reaction and disease-free survival in primary HER2 positive breast cancer [43].

## 3. Tumor Microenvironment in TNBC

Cells in the surrounding area of the tumor affect tumor growth, proliferation and migration to varying degrees in different types of breast cancer. TNBC has been shown to have a unique immunogenic TME, characterized by an important immunosuppressive landscape.

### 3.1. Cancer-Associated Fibroblasts (CAFs) in TNBC

CAFs have a prominent role in reducing antitumor immunity, enhancing tumor cell proliferation and invasion, promoting tumor cell neoangiogenesis, remodeling the extracellular matrix (EMC), and contributing to the development of an immunosuppressive microenvironment [11]. Studies with the TNBC cell lines MDA-MB-231 and BT-549 cells proved increased in vitro migration by CAFs autophagy [44]. CAFs have been demonstrated to promote TNBC development by TGF-β activation [45], and by inducing a high interleukin expression level in co-culture experiments [46]. More myeloid cells and fibroblasts are recruited by myeloid-cells-activated CAFs in TNBC [47]. If the enrichment of CAF-S1 is higher than CAF-S4 among the four CAF subtypes, the infiltration of CD8+ T cells is lower and TNBC is more aggressive [48].

### 3.2. Tumor-Infiltrating Lymphocytes (TILs) in TNBC

Infiltration of TILs, particularly in the stroma of the TNBC, has been demonstrated to be higher when compared to other subtypes of breast cancer [49,50]. The increased level of TIL is associated with an improved clinical outcome in TNBC patients. Accordingly, a recent meta-analysis demonstrated that high TIL levels are associated with a better survival outcome in TNBC, and are considered to be a good prognostic factor for this cancer subtype [51]. CD8+ T lymphocyte infiltration is associated with an improved clinical outcome and increased survival in TNBC patients [52]. Overall, hormone receptor-negative breast cancer has shown a higher activity of CD8+ T lymphocytes [53]. In TNBC, immunosuppressive Tregs constrain the activation of CD8+ and CD4+ T lymphocytes, preventing the patient’s immune response [54]. Increased Treg concentration and a higher infiltration of FOXP3+ (forkhead box P3) were described in the TNBC microenvironment, causing a reduction in the autoantigen immune response and predicting poor prognosis [55,56].

### 3.3. Tumor-Associated Macrophages (TAMs) in TNBC

M2-type TAMs are upregulated in TNBC compared to other subtypes of breast cancer. It has been shown that TNBC cells secrete more granulocyte colony-stimulating factor (G-CSF), which promote the conversion of M1-type macrophages to M2-type, thereby helping tumor development [57,58]. Importantly, TAMs also regulate the expression of programmed death ligands (PD-1/PD-L1) in the TNBC tumor environment initialization [58], which suggests that the expression of TAMs is closely associated with poor prognosis in cancer patients. It is worth nothing that TAMs secrete cytokines such as TGF-β and IL-13, which inhibit the proliferation and differentiation of lymphocytes, including lymphocytes activating killer (LAK) cells, NK cells, and cytotoxic T lymphocytes [11].

### 3.4. CAAs, TANs and NKs in TNBC

CAAs have a more pronounced effect on tumor growth and invasion in TNBC [59,60]. Additionally, TANs seem to promote tumor proliferation, migration and metastasis and inhibited anti-tumor immunity in TNBC. In addition, secreted G-CSF by TNBC cells activates TANs to promote angiogenesis and improve tumor cell infiltration [61]. Surprisingly, immature NK cells promote progression in TNBC patients through Wnt (Wingless and Int-1) signaling [62].

Overall, TILs, TAMs, CAFs, and CAA are over-regulated or over-infiltrated in TNBC stroma, of which only TILs show a positive correlation with prognosis; the others correlate negatively [11]. Identifying compounds specific to the TNBC microenvironment as biomarkers can be useful in determining personalized therapeutic strategies and/or predicting the likelihood of tumor recurrence in TNBC.

## 4. Metabolic Pathways and Signaling Activated in TNBC

### 4.1. Glycolysis

Glycolysis is more important in TNBC (Figure 3) than in other subtypes of breast cancer, as it is characterized by increased glucose uptake and lactate secretion [63]. Key glycolytic enzymes (LDH-lactate dehydrogenase) and transporters (GLUT-glucose transporter; MCT-monocarboxylate transporter) are also upregulated. GLUT4 knockdown reduces glucose uptake and lactate release, thus redirecting glycolytic flux to oxidative phosphorylation (OXPHOS), leading to a decrease in cell proliferation and viability under hypoxia [64]. LDH-A and LDH-B levels are associated with poor clinical outcomes in TNBC patients [65]. MCT1 and MCT4 isoforms play a role in lactate extrusion and acidification of TME, and are specifically upregulated in TNBC cells [66].

### 4.2. OXPHOS

Although OXPHOS produces more ATP than glycolysis, glycolysis remains the main bioenergetic source in most tumors, including TNBC [67]. In the absence of oxygen, electrons escape from the electron transport chain and maintain higher ROS levels that cause ROS-mediated DNA damage in tumors [68]. Culturing TNBC cell lines under hypoxic conditions shifts the metabolic processes towards glycolysis by downregulating OXPHOS [69]. This process is regulated by an enzyme called fructose-1,6-bisphosphatase (FBP), which inhibits the activity of hypoxia inducible factor 1 subunit alpha (HIF-1α) and promotes the transcription of the GLUT1, LDHA, and PDK194 genes involved in glycolysis. At the same time, MYC overexpressed in TNBC cells promotes OXPHOS and ROS production, resulting in the production of breast cancer stem cells (BCSCs) and resistance to chemotherapy [70].

In the hypoxic tumor microenvironment, HIF-1α is activated through the mTOR (mammalian target of rapamycin), NF-κB (nuclear factor kappa B), and JAK (Janus kinase)-STAT signaling pathways by receptors on the plasma membrane (e.g., TCR-T cell receptor, GFR-growth factor receptor, IL-6R and TLR-Toll-like receptor) [71]. Upregulated HIF-1α combines with HIF-1β to form a dimer in the nucleus. The transcription of several genes, such as VEGF, erythropoietin (EPO), and STAT3, is involved in the adaptation of cells to hypoxic stress.

Under physiological conditions, HIF-1α and AMP-activated protein kinase (AMPK) activate the process of glycolysis and OXPHOS (Figure 4) [72]. HIF-1α is degraded by prolyl hydroxylase domain 2 (PHD2) under the action of alpha-ketoglutarate (α-KG) and cysteine. In TNBC, α-KG activity is reduced due to decreased transketolase (TKT) activity, and cysteine levels are reduced due to altered xCT-cystine/glutamate antiporter (also known as SLC7A11) activity. This contributes to normoxic activation of HIF-1α signaling, which triggers aerobic glycolysis by upregulation of glycolysis-related enzymes and transporters [73]. TNBC cells are assisted by glycolysis to proliferate rapidly but also require OXPHOS, which produces ATP more efficiently under bioenergetic stress. Mitochondrial biogenesis, induced by activated AMPK and MYC pathways and enzymes involved in glutaminolysis and fatty acid oxidation (FAO), play a key role in the choice between glycolysis and OXPHOS [74]. Furthermore, AMPK signaling promotes epithelial-mesenchymal transition (EMT) by regulating lipid metabolism, and recruits immunosuppressive myeloid-derived suppressor cells by increasing cytokine secretion.

### 4.3. Amino Acid Metabolism

#### 4.3.1. Glutamine

Glutamine is a non-essential amino acid in the body; nevertheless, it plays an essential role in supporting biosynthesis and energy generation in TNBC by entering the TCA cycle following transformation [75]. Transporters responsible for importing glutamine to the cells include ASCT2 (alanine, serine, cysteine-preferring transporter 2) and LAT1 (L-type amino acid transporter 1) that are overexpressed in TNBC [76] (Figure 4). Glutaminase (GLS), which deaminates intracellular glutamine to glutamate in the process of glutaminolysis, is also overexpressed in TNBC [77]. The elevated level of glutaminolysis in TNBC [78] provides an opportunity to develop small molecule inhibitors of GLS such as CB-839, BPTES, and compound 968 [79]. Glutamate can be converted to α-KG in the mitochondria by transaminase or glutamate dehydrogenase (GLUD) and then enter the TCA cycle. Overexpression of these transaminases (GPT2-glutamic-pyruvic transaminase 2, PSAT1-phosphoserine aminotransferase 1, and GOT2-glutamic-oxaloacetic transaminase 2) supports cell proliferation by increasing aspartate and α-KG production in TNBC. BRCA1 (breast cancer type 1) protein transcriptionally suppresses GOT2 expression, but this mechanism is impaired by BRCA1 deficiency often seen in TNBC [80].

The increased glutaminolysis in TNBC, and consequently the high level of glutamate, indirectly promotes the accumulation of cystine through the action of the xCT antiporter (Figure 4). Under normal conditions, extracellular cystine enters the cell at the same time as intracellular glutamate is pumped out, and supports PHD2 which hydroxylates HIF-1α for degradation, thus inhibiting HIF-1α signaling. Upregulated xCT is shown in one-third of TNBC, and is indispensable for GSH synthesis and the maintenance of cancer stem cells (CSCs) [81]. xCT is stabilized at the cell membrane by the CD44 variant (CD44v), a marker of CSCs [82]. The transmembrane glycoprotein mucin-1 (MUC1), which is bound directly to the intracellular domain of CD44v, further promotes the stability of xCT in TNBC [83]. As increased glutamate secretion decreases cystine uptake into the cell. TNBC carcinogenesis may be triggered by the HIF-1α pathway via PHD2 deactivation [84]. Metabotropic glutamate receptors (mGluR) on the membrane of TNBC are also induced by an increased level of secreted glutamate, thus promoting tumor growth, angiogenesis, and inhibiting inflammation via the mGluR-signaling pathway.

#### 4.3.2. Serine and Glycine

The glycolytic carbon flux is directed from 3-phosphoglycerate towards de novo serine and glycine biosynthesis via the serine synthetic pathway (Figure 3). The pathway has several metabolic benefits, such as limiting the production of ATP and serine through the use of monosaccharide metabolism and the formation of NADPH. Additionally, this pathway produces α-KG, an intermediary in the TCA cycle, from glutamate. Phosphoglycerate dehydrogenase (PHGDH), a key enzyme in serine synthesis, is overexpressed in TNBC and basal-like breast cancer, and leads to oncogenesis through disrupting morphogenesis in breast epithelial cells and inducing phenotypic changes [85]. Reduced cell proliferation, serine synthesis, and lower levels of α-KG are caused by the suppression of PHGDH and PSAT1. In addition, the overexpression of PHGDH and PSAT have been associated with a poor clinical outcome and more aggressive phenotypic features [86].

#### 4.3.3. Tryptophan

Most tryptophan catabolism via indoleamine 2,3-dioxygenase (IDO), tryptophan 2,3-dioxygenase (TDO) [87], or tryptophan hydroxytryptamine 1 (TPH1) enzymes occurs through the kynurenine pathway (Figure 4). The end products of catabolism are kynurenine and 5-hydroxytryptamine (5-HT; also called serotonin), which can significantly modulate the immune response and oncogenic signaling in TNBC through the aberrant expression of catalyzing enzymes triggered by T cell-derived interferon gamma (IFN-γ) [88]. Overexpression of TDO2 in TNBC cells makes TNBC more resistant to programmed cell death in a NF-κB-dependent manner, and promotes TNBC proliferation, invasion, and metastatic ability [89]. Additionally, 5-HT increases the expression of TPH1 and VEGF in TNBC cell invasion and proliferation via activation of the 5-HT7 receptor.

#### 4.3.4. Arginine

Arginine is important for the growth of TNBC because it produces ornithine and nitric oxide (NO). Ornithine is synthesized in cancer cells by arginase 2 (ARG2) during S/G2/M phases only, and by the ornithine aminotransferase (OAT) in normal cells. Cancer cell growth is markedly reduced when ARG2 expression is knocked down in basal-like breast cancer [90]. Breast cancer development can be blocked by rosuvastatin by inhibiting arginase enzymatic activity and reducing the level of ornithine and polyamine [91]. Arginine is also used for NO synthesis, a molecule that has multiple functions in the cell. High activity of inducible nitric oxide synthase (iNOS), which produces NO directly, often leads to poor survival rates in TNBC patients. This happens because increased production of NO activates the EGFR pathway, and therefore various oncogenic signaling pathways including c-MYC, AKT, and β-catenin. EMT, chemoresistance, and invasion ability are enhanced by NO signaling upregulating the stem cell marker CD44 and other basal-like breast cancer-specific proteins [92].

The metabolism of asparagine, methionine, and glutamine are essential in regulating the growth and metastasis of TNBC cancer cells. Dietary restrictions or medications that impede their synthesis can have tumor-suppressive effects [93,94,95].

### 4.4. Lipid Metabolism

Another efficient way for tumor cells to produce energy comes from fatty acid oxidation (FAO). A higher level of fatty acid oxidation (FAO) and downregulation of fatty acid synthesis (FAS) were described in TNBC. In FAO, acyl-CoA is first formed by the acylation of fatty acid, which is then transferred to the mitochondria via the carnitine palmitoyl transferase (CPT) (Figure 5). Since CPT activity is upregulated in MYC-overexpressing TNBC, there is an increased bioenergetic requirement for FAO. Higher levels of FAO and ATP production activate the Src oncoprotein and increase cellular resistance to metabolic stress induced by hypoxia, glucose deprivation, or mTOR inhibition [96]. In turn, disruption of CPT1 leads to in vivo tumor growth and reduced metastasis by abolishing Src activation [97]. TNBC metastasis is further facilitated by upregulated peroxisome proliferator-activated receptor gamma coactivator 1 alpha (PGC1-α), a key regulator of mitochondrial biogenesis and respiration, which is upregulated by elevated FAO [98]. This activates FAO under metabolic stress to maintain energy balance and promote cell viability [99]. In contrast, fatty acid synthesis (FAS) and lipogenic enzymes are downregulated in TNBC compared to other subtypes [75]. Acetyl-CoA is irreversibly catalyzed by acetyl-CoA carboxylase (ACC) to malonyl-CoA, which inhibits CPT1 and FAO. In TNBC, regulation of ACC activity decreases malonyl-CoA generation and increases FAO flux. Through AMPK signaling, ACC1 can be inactivated by TGF-ß or leptin, thereby increasing cellular acetyl-CoA levels, contributing to the acetylation of Smad2 (SMAD Family Member 2) and ultimately to the development of EMT programs and metastasis [100] (Figure 5).

Fatty acid cells supplying bioenergy may be derived also from triglycerides in circulating lipoprotein particles. Upregulation of lipoprotein lipase (LDL), which hydrolyzes triglyceride into fatty acids and proteins in the fatty acid binding protein (FABP) family, are associated with a poor clinical outcome in TNBC [101]. Higher uptake and utilization of cholesterol by elevated activity of low-density lipoprotein (LDL), caveolin-1, and cholesterol acyltransferase 1 (ACAT1) were also found in TNBC, leading to elevated cholesterol biosynthesis in the mevalonate pathway [102]. NF-κB signaling is also activated and supports EMT programs by the upregulation of aldo–keto reductase 1 member B1 (AKR1B1) by the positive feedback of Twist2 activation [103] (Figure 5).

## 5. Epigenetic Changes in TNBC

While genomic changes of TNBC include mutations, copy number variations, and genetic rearrangements, TNBC is also characterized by epigenetic signatures, including changes in DNA methylation and histone remodeling [104]. These epigenetic modifications either silence or activate genes that were previously described in TNBC. In addition, recent evidence supports epigenetic regulation of extracellular matrix alterations in TNBC. This reinforces the importance of epigenetic mechanisms, which play a role in the pathogenesis, maintenance, and resistance to therapy [105]. For example, TNBC metastasis could be inhibited by Fbxo22 (F-box protein 22) through ubiquitin modification of KDM5A (Lysine Demethylase 5A) and regulation of H3K4me3 demethylation [106]. Genome- and transcriptome-wide analyses in TNBC show the importance of histone H3 proline 16 hydroxylation (H3P16oh) in the regulation of mammalian gene expression, and DKK1 (Dickkopf-related protein 1), a negative regulator of the Wnt pathway, is repressed via the EGLN2 (Egl-9 family hypoxia inducible factor 2)-H3P16oh-KDM5A pathway to promote Wnt/β-catenin signaling [107]. An increased level of the long non-coding RNA LINC01559 in TNBC is associated with tumor growth and lung metastasis in a xenograft model via increasing the expression of oncogenes [108].

Anti-epigenetic drugs have shown promising results in the pre-clinical and clinical setting in breast cancer [109]. Natural compounds such as thymoquinone, regorafenib, Fangji Huangqi Decoction, saikosaponin A, and Trametes robiniophila Murr extracts were reported in the literature as potential antitumor agents in TNBC [104]. Combination therapies would probably work better in controlling cancer growth and metastasis; for example, a study of neoadjuvant paclitaxel and epigenetic drugs led to a suppression of cancer progression by inhibiting post-treatment regrowth of TNBC cells [110]. The ongoing advances in epigenetic research will help to expand the armamentarium of treatments for TNBC [111].

## 6. Diagnostic and Therapeutic Possibilities Linked to Tumor Metabolism in TNBC

TNBC is the most aggressive and difficult to treat subtype of breast cancer. The overall biologic heterogeneity that characterizes TNBC can greatly complicate the identification of effective therapies with lasting efficacy, making TNBC difficult to treat with current therapies [112]. High metastatic potential and poor prognosis remains an important clinical feature of this breast cancer subtype. Among cancer hallmarks, the presence of an immunosuppressive TME and a unique metabolic reprogramming characterizing TNBC could effectively provide opportunities for novel therapeutic strategies.

A higher overall pathological response to neoadjuvant chemotherapy in pretreatment biopsies and better survival are predicted by higher TIL levels according to recent meta-analyses [113,114]. In more than 100 patients with TNBC, higher CD8+/FOXP3+ ratios after neoadjuvant chemotherapy were associated with a better prognosis [115]. These findings are also supported by a study investigating the relationship between TILs, immune response regulators, and the glycolytic TME in a large number of TNBC patients [116]. A higher expression of TILs was associated with better survival, although expression of the immune markers PD-L1, FOXP3, and CD163 was significantly associated with reduced overall survival. The authors raised the possibility of further subclassifying TNBC patients according to the expression of these determinants and MCT4 to predict survival. Expressions of immune checkpoints PD-1 with LAG-3 were revealed to be potentially determining factors to clinically evaluate early stage TNBC patients [117,118]. Higher TILs at diagnosis were associated with decreased distant recurrence rates in primary TNBC and increased trastuzumab benefit in HER2+ disease in the FinHER trial [119]. Another recent study has demonstrated a positive prognostic role for increased TIL levels in a study of 247 TNBC patients receiving neoadjuvant treatment containing CMF (cyclophosphamide, methotrexate and fluorouracil), or a combination with anthracyclines (cyclophosphamide and doxorubicin) [120]. Ex vivo treatment of TILs with pentoxifylline in a TNBC mouse model reduced the proportion of Tregs in conventional IL-2-mediated TIL expansion and improved the anti-tumor immune response by altering the cytokine balance of TILs [121]. These results provide strong evidence that stromal TILs can be used both as a predictive marker of a response to chemotherapy and as a prognostic marker. TILs are often inhibited by the presence of immunosuppressive molecules expressed in TNBC, and targeting such molecules relieving T-cell inhibition could elicit a relevant anti-tumor response.

The important role of CAF cells in tumor stroma formation potentially raises the possibility of using upregulated genes as biomarkers. Expression of MCT4 in the tumor stroma, which represents a glycolytic tumor environment, is associated with poor prognosis in TNBC patients [116]. Potential CAFs linked targets [122] could be included on the inhibition of CAF transition from inactive to active form [123]; the utilization of vitamin D to revert CAFs to an inactive state, as has been shown in colorectal cancer and pancreatic cancer [124,125]; the reduction of CAFs by CAR-T cell therapy [126], cancer vaccine [127] or monoclonal antibody therapy [128]; the negation of CAF tumorigenic functions through inhibition of EMT, stem cell formation, and metastasis [126]; the reduction of the immunosuppressive functions of CAFs to increase T cell access to tumor cells and their sensitivity to therapeutic agents [128]; and the inhibition of CAF-derived substances (chemokines, cytokines, exosomes, miRNAs, and ECM proteins) [129]. Although to date no specific study was performed targeting CAFs in TNBC, several of the listed pathways offer promising opportunities for developing therapeutic agents in TNBC as well.

It has been shown that, mainly through the substances they secrete, adipocytes can promote tumor progression, enhance immunosuppressive TME and compromise therapeutic efficacy. When a protein trap against CC motif chemokine ligand 2 (CCL2), produced primarily by CAAs, was delivered into the TME using nanoparticles, enhanced therapeutic efficacy and significant tumor growth inhibition were observed [130]. In addition, T cell infiltration was increased and the populations of immunosuppressive M2 macrophages and myeloid-derived suppressor cells (MDSCs) were reduced. Nanotechnology-based therapies, which have gained ground recently, represent a promising approach to target TME and improve treatment outcomes in TNBC patients, which may lead to improved survival and quality of life [131].

The recruitment of TAMs in the TME was affected by omega-3 docosahexaenoyl ethanolamide, which in turn affected tumor progression and macrophage recruitment in TNBC cells by reducing CCL5 secretion [132]. In another study, tinengotinib (TT-00420), which strongly inhibited Aurora A/B, FGFR1/2/3, VEGFR, JAK1/2, and CSF-1R in biochemical assays, specifically inhibited the proliferation in all subtypes of TNBC in vitro and in vivo, while leaving luminal breast cancer cells intact. The studies showed that the potential mechanism of action was primarily through the inhibition of Aurora A or B kinase activity, and resulted in reduced TAM infiltration. A phase I trial of tinengotinib has been completed, and showed promise as a combinatorial inhibitory mechanism for the treatment of TNBC [133]. Targeting Notch, IL1β, or CCL2 may reduce TAM recruitment and resistance to immune checkpoint inhibitors, shedding light on the potential of combination immunotherapy in TNBC [134].

Metabolic reprogramming is also involved in intracellular energy production processes. In TNBC, the major glycolytic transporters and enzymes significantly upregulate glycolysis and its downstream pathways (Figure 3). NADPH production is enhanced by serine synthesis and the pentose phosphate pathway (PPP), whereas protein glycosylation is upregulated by the hexosamine biosynthetic pathway (HBP). Fatty acid oxidation (FAO), glutaminolysis, oxidative phosphorylation (OXPHOS), and cystine uptake are also overactivated in TNBC due to increased bioenergetic demand. These processes allow cancer cells to rapidly produce energy and biosynthetic intermediates to support their rapid growth and proliferation. There are a number of therapeutic strategies to target the reprogrammed energy production processes in TNBC. One strategy is the use of glycolysis inhibitors such as 2-deoxyglucose (2-DG) [135,136] and the use of the combination cisplatin-tegafur-lonidamine [137], which act by interfering with glycolysis and sensitizing tumor cells to additional agents. Although glycolytic inhibitors have shown promise in preclinical studies, they have not yet shown significant efficacy in clinical trials. This is probably due to these agents’ limited specificity and potential toxicity.

The inhibition of glycolysis in TNBC has dual consequences: it suppresses tumor cell aggressiveness and the cancer stem cell (CSC) phenotype [138], while shifting metabolic processes towards OXPHOS, thus shifting energy metabolism from the mitochondria to the cytoplasm. ROS is known to induce a shift of breast cancer stem cells (BCSC) from a mesenchymal state (high ALDH levels) to an epithelial state (high CD24-CD44+ expression) via the AMPK-HIF-1α pathway. The latter exhibits high OXPHOS activity and antioxidant protection [139], which is also observed in brain or lung metastases of TNBC [140,141]. Following chemotherapy, tumors become highly sensitive to OXPHOS inhibitors [142]. It has been reported that a specific hybrid phenotype of TNBC exists in which glycolysis and OXPHOS may be present simultaneously. This results in a higher metabolic flexibility to metabolic drugs [72], thus leading to a more proliferative and aggressive spread. A dual attack of these two pathways may be an effective strategy to promote cancer cell death through the inhibition of bioenergetic processes [139].

Several key signaling pathways are involved in the metabolic reprogramming of TNBC cells. The PI3K/AKT/mTOR pathway is physiologically involved in cell metabolism, growth, proliferation, and apoptosis through the activation of tyrosine kinase receptors (RTK) and G-protein-coupled receptors [143]. AKT is the central mediator of the PI3K pathway, activating more than one hundred substrates including mTOR, while PTEN (phosphatase and tensin homolog) is the main negative regulator of PI3K signaling [144]. Changes in the PI3K/AKT/mTOR pathway during cancer development are mainly due to mutations in PIK3CA and AKT, overexpression of RTKs or loss of PTEN [145]. Combined activating mutations in PIK3CA and AKT1 occur in 25–30% of advanced TNBC [146]. Mutations and copy number changes in the TP53 and PIK3CA/AKT genes in plasma may be important markers of TNBC development, progression, metastasis, and in the clinical follow-up [147]. The PI3K/Akt/mTOR pathway is often dysregulated in TNBC, which also promotes glycolysis by upregulating glycolytic enzymes while suppressing OXPHOS by inhibiting mitochondrial biogenesis and function. It has been shown that the AMPK activator AICAR (5-Aminoimidazole-4-carboxamide ribonucleotide) can be combined with rapamycin in breast cancer cells [148] and other cancers [149], and has a definite role in cancer metabolism. Glutamine deprivation generates “synthetic lethality” for cytotoxic drugs capecitabine, paclitaxel, and rapamycin in cancer cells driven by KRAS (Kirsten rat sarcoma virus) arresting the cells in S phase [150]. The selective inhibition of glutamine uptake also seems to be a promising therapeutic strategy in TNBC [151]. Several plant active compounds have been shown to strongly inhibit pathways involved in CSC in vitro and in vivo [152,153,154,155,156]. The AKT-inhibiting neoadjuvant ipatasertib sensitized TNBC to paclitaxel [157]. The combination of doxorubicin, metformin, and oxamate resulted in effective and rapid tumor growth inhibition in a xenograft model by inhibiting mTOR phosphorylation and LDH-A gene expression. The mTOR inhibitor everolimus [158] and the PI3K inhibitor buparlisib [159] have shown promising results in preclinical and early clinical studies, where they demonstrated antitumor activity and sensitivity to chemotherapy in TNBC. Synthetic flavonoids targeting aromatase inhibitors in breast cancer [160], PARP inhibitors [161], and immune checkpoint inhibitors [162], alone or in combination, seem to be promising in the treatment of TNBC patients. The HIF pathway, which is activated in response to low oxygen levels, also plays a critical role in the regulation of glycolysis in TNBC and is a promising target to overcome chemoresistance [163]. Several drugs targeting these pathways are currently in development or in clinical trials, including immunotherapy, chemotherapy and anti-angiogenic therapy (Appendix A).

## 7. Conclusions

Despite the approval of recent drugs in TNBC, many patients still show poor response to therapies, early relapse and/or drug resistance, making this breast cancer subtype still an unmet medical challenge. In this review, we described all cells and molecules present in the TNBC microenvironment that can also influence the metabolic processes inside the tumor cells, thus accelerating tumor progression. Metabolic reprogramming that characterizes TNBC involves a switch in energy production from oxidative phosphorylation to glycolysis. This shift not only provides energy for tumor growth but also leads to the increased production of lactate, which can create an acidic and hostile tumor microenvironment for immune cells, promoting tumor progression and metastasis. To date, a number of drugs and combinations have been tested against these metabolic signaling pathways to inhibit tumor cell aggressiveness, angiogenesis, and metastasis, but breakthrough results are still awaited. A comprehensive understanding of metabolic and TME interactions, related signaling, and mechanism of interactions are needed to identify novel and better treatment options in TNBC.

## Figures and Tables

**Figure 1 ijms-24-06945-f001:**
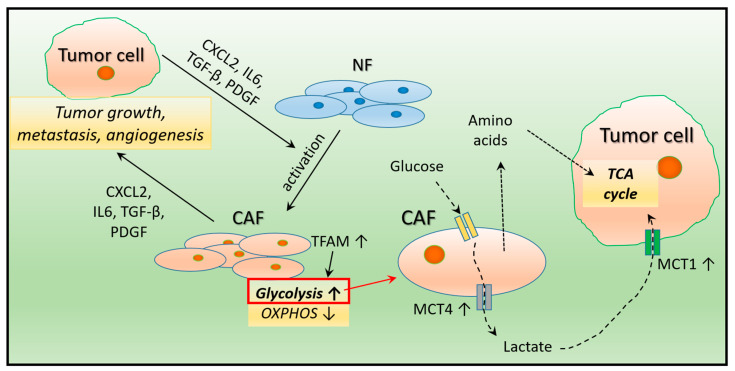
The effect of tumor cells on fibroblast reprogramming. The substances secreted by tumor cells convert normal fibroblasts in the environment into cancer-associated fibroblasts (CAF), which support tumor cell growth, metastasis, and angiogenesis. In CAF cells with increased glycolysis, uptaken glucose is converted to lactate, which is driven by increased efflux to tumor cells through higher MCT4 (monocarboxylate transporter 4) levels. In turn, higher MCT1 levels in tumor cells allow for increased lactate uptake, which is then utilized in the mitochondria via the TCA (tricarboxylic acid) cycle. The amino acids produced by CAF cells offer tumor cells another opportunity for energy utilization. NF: normal fibroblast.

**Figure 2 ijms-24-06945-f002:**
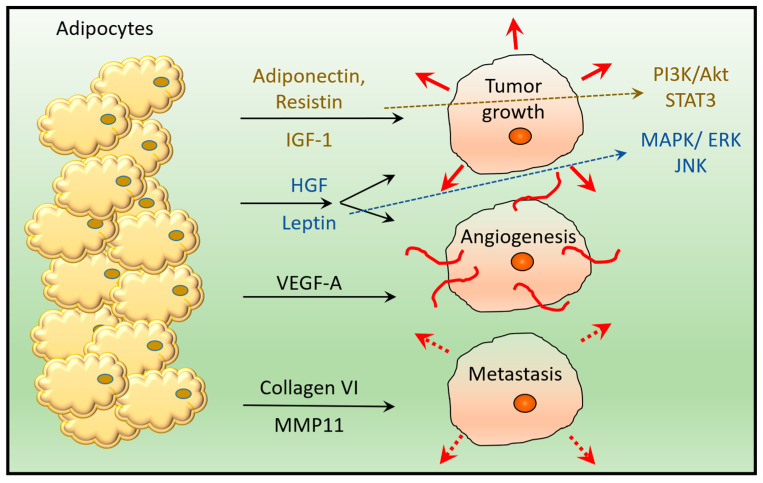
Compounds secreted by adipocytes support tumor growth, angiogenesis and metastasis. Adipocyte-secreted adiponectin, resistin, IGF-1 (insulin-like growth factor 1), and HGF (hepatocyte growth factor) induce tumor growth via activation of the PI3K/Akt (phosphoinositide 3-kinase/protein kinase B) and STAT3 (signal transducer and activator of transcription protein 3) pathway. IGF-1 and HGF also activate the MAPK/ERK (mitogen-activated protein kinase/extracellular signal-regulated kinase) pathway, together with leptin. HGF, Leptin and VEGF-A (vascular endothelial growth factor A) are involved in angiogenesis, while secreted collagen VI and MMP11 (matrix metalloproteinase 11) play a role in tumor metastasis.

**Figure 3 ijms-24-06945-f003:**
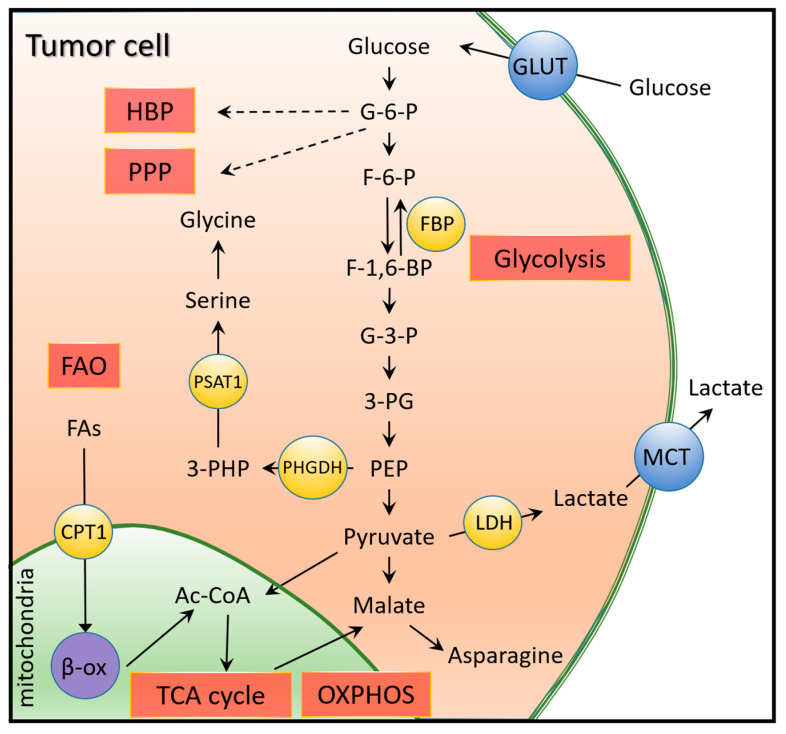
Glycolysis and its relationship with other metabolic pathways in TNBC. Due to higher glucose uptake in TNBC, the pyruvate produced from glucose enters the TCA cycle in mitochondria as Ac-CoA (acetyl coenzyme A). The malate that exits or is converted from pyruvate forms asparagine, whose high expression is associated with poor prognosis in TNBC. The high LDH (lactate dehydrogenase) expression allows elevated lactate efflux, which acidifies extracellular matrix and promotes tumor aggressiveness. Upregulated protein glycosylation via the hexosamine biosynthetic pathway (HBP) and increased NADPH production is described by upregulated pentose phosphate pathway (PPP) and serine synthesis. Additionally, higher fatty acid synthesis (FAO) characterized in TNBC also promotes the energy utilization of tumor cells.

**Figure 4 ijms-24-06945-f004:**
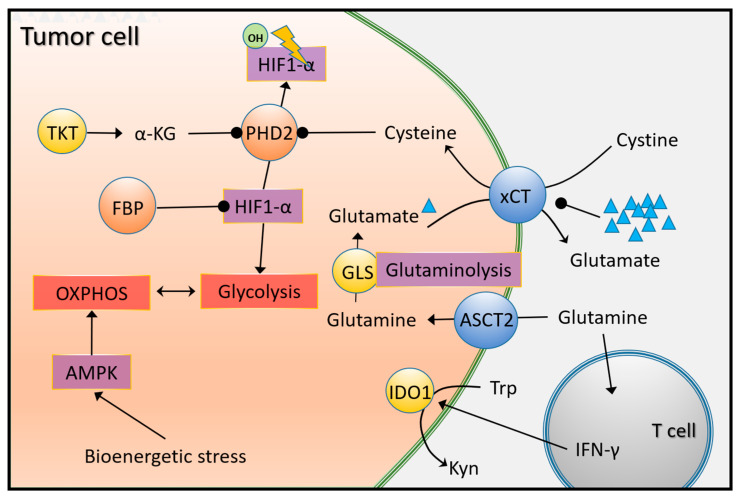
HIF-1α-, glutamine-, and tryptophan-dependent metabolic reprogramming in TNBC. Under physiological conditions, PHD2 (prolyl hydroxylase domain 2) hydroxylates and degrades HIF-1α (hypoxia inducible factor 1 subunit alpha) in an α-KG (alpha-ketoglutarate) and cysteine-dependent manner, thereby inhibiting aerobe glycolysis. In TNBC, PHD2 is inhibited by the cystine/glutamate antiporter xCT and TKT (transketolase), thereby upregulating enzymes and transporters involved in glycolysis. In addition to glycolysis, the entry of amino acids into the TCA cycle also leads to energy gains. In TNBC, the levels of the ASCT2 transporter (responsible for glutamine uptake) and the enzyme GLS, which converts it to glutamate, are upregulated. Meanwhile, catabolism of tryptophan (Trp) is facilitated by overexpressed IDO1. The resulting kynurenine (Kyn) is involved in impairing the immune response and facilitating oncogenic signaling through T cell-derived IFN-γ (interferon γ).

**Figure 5 ijms-24-06945-f005:**
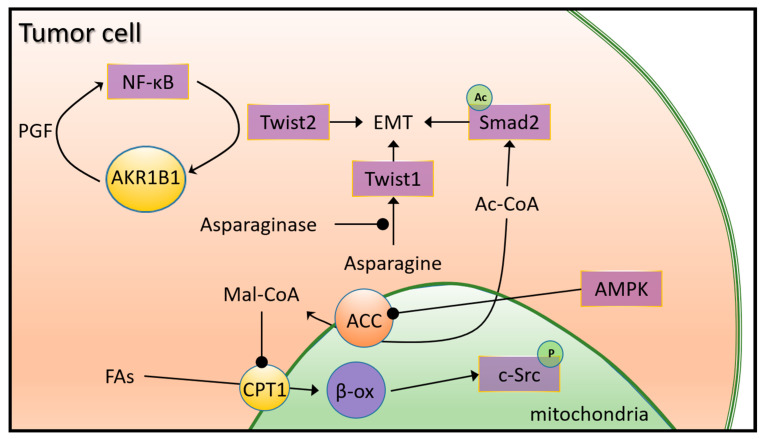
Lipid metabolic reprogramming in TNBC. The key lipogenic enzyme acetyl-CoA carboxylase (ACC) irreversibly catalyzes acetyl-CoA to malonyl-CoA in the mitochondrial membrane, which is an inhibitor of CPT1 and FAO. FAO flux in TNBC is promoted by downregulated ACC, which may also be caused by AMPK (AMP-activated protein kinase) signaling. The resulting increase in cellular acetyl-CoA promotes EMT-inducing Smad2 (SMAD Family Member 2) acetylation and ultimately, activation of EMT pathways and metastasis. Upregulated AKR1B1 in TNBC activates the NF-κB (nuclear factor kappa B) signaling through its accumulated metabolite PGF, which leads to the overexpression of Twist2 and enhances cancer stem cell (CSC) properties in TNBC. Higher asparagine levels are associated with less metastatic potential in TNBC by activation of Twist1 molecule through downregulation of the inhibitor asparaginase.

## Data Availability

Not applicable.

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
