# Peer review of "Therapeutic Potential of Tumor Metabolic Reprogramming in Triple-Negative Breast Cancer"

_ijms, 2023, doi:10.3390/ijms24086945_

Round 1

Reviewer 1 Report

1. There is no introduction. Please add an introduction to the article. This introduction should include breast cancer, TNBC, statistics and therapies, metabolism dysregulations in TNBC, etc with recent references.

2. 3. Metabolic pathways and signaling activated in TNBC has to be the second part with recent data

3. Write about epigenetics in TNBC in a different section

4. Write about cancer hallmarks and effects of metabokism in a seperate section

5. Divide subsections (amino acid metabolism, lipid and glucose

6. Write about major signaling components like HIF, mTOR, AKT in TNBC and their roles in metabolism

7. See this recent review in an MDPI journal (https://www.mdpi.com/2218-1989/13/3/345). Cite it

8. Write about different enzyme inhibitors

9. Take section 1 to last. The micro-environment 

Author Response

Reviewer 1#.

  • There is no introduction. Please add an introduction to the article. This introduction should include breast cancer, TNBC, statistics and therapies, metabolism dysregulations in TNBC, etc. with recent references.

We thank the Reviewer for this important remark. We have extended the manuscript with an introduction section citing recent references.

  • Metabolic pathways and signaling activated in TNBC has to be the second part with recent data.
  • Take section 1 to last. The micro-environment.

We thank the Reviewer for these suggestions. We have explored the possibility of moving the sections by rethinking the original message and the structure of this review. We consider the sections to be sequential in their current placement, reflecting our original aim. Yet, following your suggestion, we have inserted additional recent data in these paragraphs to enhance the strength of the review.

  • Write about epigenetics in TNBC in a different section

We thank the Reviewer to pay attention to this relevant issue. We have extended the manuscript with a new section.

  • Write about cancer hallmarks and effects of metabolism in a separate section.

Thanks for this relevant suggestion. We integrated relevant findings about cancer hallmarks and effects of metabolism into the review.

 Divide subsections (amino acid metabolism, lipid and glucose)

Thanks for the suggestion. We have rearranged this section and several other parts of the manuscript by dividing them into subsections.

  • Write about major signaling components like HIF, mTOR, AKT in TNBC and their roles in metabolism

We thank the Reviewer for bringing this important issue to our attention. We have expanded the manuscript with relevant findings and recent references about these components.

  • See this recent review in an MDPI journal (https://www.mdpi.com/2218-1989/13/3/345). Cite it

We thank the Reviewer for drawing our attention to this article and have added it to our manuscript.

  • Write about different enzyme inhibitors

We have extended the Diagnostic and therapeutic possibilities linked to tumor metabolism in TNBC section with relevant findings as suggested.

Reviewer 2 Report

A timely review article by Dr. Győrffy and the group elaborates on the role of metabolic reprogramming in breast cancer and also discusses its therapeutic implications of it. This is a very well-written review article that has a translational aspect. However, a few things must be addressed before it is ready for acceptance. They are as follows:

1. It has been shown that AMPK activator- AICAR can be combined with Rapamycin in breast cancer cells (PMID: 26323019) and other cancers (PMID: 30107094). The authors should discuss this point which has a definite role in cancer metabolism.

2. It has been discussed how metabolic checkpoints play a role in cancer therapeutics (PMID: 26682255). 

The authors should add a few lines discussing this point in the context of triple-negative breast cancer. 

3. Add a table with all current clinical trials dealing with metabolism-related breast cancer drugs. 

Author Response

Reviewer 2#. A timely review article by Dr. Győrffy and the group elaborates on the role of metabolic reprogramming in breast cancer and also discusses its therapeutic implications of it. This is a very well-written review article that has a translational aspect.

We thank for the positive remark of the Reviewer.

However, a few things must be addressed before it is ready for acceptance. They are as follows:

  • It has been shown that AMPK activator- AICAR can be combined with Rapamycin in breast cancer cells (PMID: 26323019) and other cancers (PMID: 30107094). The authors should discuss this point which has a definite role in cancer metabolism.

We thank the remark of the Reviewer and have extended the manuscript with this additional topic.

  • It has been discussed how metabolic checkpoints play a role in cancer therapeutics (PMID: 26682255). The authors should add a few lines discussing this point in the context of triple-negative breast cancer.

We thank the remark and have added the reference to the manuscript regarding this issue as suggested.

  • Add a table with all current clinical trials dealing with metabolism-related breast cancer drugs.

We thank the Reviewer for the highly relevant remarks. We have extended the manuscript with an additional table with ongoing trials on metabolism-related breast cancer drugs (Table S1.)

Reviewer 3 Report

The manuscript entitled "Therapeutic potential of tumor metabolic reprogramming in triple-negative breast cancer" in which the authors discussed the mechanisms in tumor metabolic reprogramming linking these changes to potential targetable molecular mechanisms to generate new physical science-inspired clinical translational insights for the cure of Triple-negative breast cancer (TNBC). The work is understandable and the topic is important. The scientific narrative is well structured and flows naturally from one idea to the next. The figures are interesting.

However, this paper suffers from few shortcomings that if modified would make the manuscript very suitable for publication in International Journal of Molecular Sciences.

Shortcomings:

1-      Please define (CXCL2), cytokines (IL-6), growth factors (TGF), line 45,46 in the first mention then write the abbreviation after that.

2-      Please define the abbreviated words in figure 3 legend e.g. GLUT, LDH, OXPHOS, ….etc.

3-      The authors write “Several drugs targeting these pathways are currently in development or in clinical trials, including immunotherapy, chemotherapy, anti-angiogenic therapy [126,127].”. Please discuss in brief the examples of these therapies showing their effects and relation to target the different mentioned pathways in this manuscript for the treatment

Author Response

Reviewer 3#: The manuscript entitled "Therapeutic potential of tumor metabolic reprogramming in triple-negative breast cancer" in which the authors discussed the mechanisms in tumor metabolic reprogramming linking these changes to potential targetable molecular mechanisms to generate new physical science-inspired clinical translational insights for the cure of Triple-negative breast cancer (TNBC). The work is understandable and the topic is important. The scientific narrative is well structured and flows naturally from one idea to the next. The figures are interesting.

We thank for the positive remarks of the Reviewer.

However, this paper suffers from few shortcomings that if modified would make the manuscript very suitable for publication in International Journal of Molecular Sciences.

  • Please define (CXCL2), cytokines (IL-6), growth factors (TGF), line 45,46 in the first mention then write the abbreviation after that. Please define the abbreviated words in figure 3 legend e.g. GLUT, LDH, OXPHOS, ….etc.

We thank the remarks of the Reviewer. We have defined the names in the manuscript after their first mentioning.

  • The authors write “Several drugs targeting these pathways are currently in development or in clinical trials, including immunotherapy, chemotherapy, anti-angiogenic therapy [126,127].” Please discuss in brief the examples of these therapies showing their effects and relation to target the different mentioned pathways in this manuscript for the treatment

We thank the Reviewer for the highly relevant remark. We have extended the manuscript with an additional table with ongoing clinical trials in relation to breast cancer metabolism (Table S1).

Round 2

Reviewer 2 Report

All concerns have been addressed, ready for acceptance. 

Reviewer 3 Report

The manuscript entitled "Therapeutic potential of tumor metabolic reprogramming in triple-negative breast cancer" in which the authors discussed the mechanisms in tumor metabolic reprogramming linking these changes to potential targetable molecular mechanisms to generate new physical science-inspired clinical translational insights for the cure of Triple-negative breast cancer (TNBC). The work is understandable and the topic is important. The scientific narrative is well structured and flows naturally from one idea to the next. The figures are interesting.

The revised manuscript is improved compared to prior revision. All my comments were adequately answered and explained by the authors. Therefore, I consider that the revised manuscript is acceptable and suitable for publication in International Journal of Molecular Sciences.
